# Host microbiome depletion attenuates biofluid metabolite responses following radiation exposure

Evan L. Pannkuk[1,2,3]*, Igor Shuryak[4], Anika Kot[1], Lorreta Yun-Tien Lin[1], Heng-Hong Li[1,2], Albert J. Fornace, Jr.[1,2,3]

1 Department of Oncology, Lombardi Comprehensive Cancer Center, Georgetown University Medical Center, Washington, DC, United States of America, 2 Department of Biochemistry and Molecular & Cellular Biology, Georgetown University Medical Center, Washington, DC, United States of America, 3 Center for Metabolomics Studies, Georgetown University, Washington, DC, United States of America, 4 Center for Radiological Research, Columbia University Irving Medical Center, New York, NY, United States of America

* elp44@georgetown.edu

**Data Availability Statement:** Data is available in the NIH data repository, Metabolomics Workbench Study ST003000 (http://dev.metabolomicsworkbench.org:22222/data/

## Abstract

Development of novel biodosimetry assays and medical countermeasures is needed to obtain a level of radiation preparedness in the event of malicious or accidental mass exposures to ionizing radiation (IR). For biodosimetry, metabolic profiling with mass spectrometry (MS) platforms has identified several small molecules in easily accessible biofluids that are promising for dose reconstruction. As our microbiome has profound effects on biofluid metabolite composition, it is of interest how variation in the host microbiome may affect metabolomics based biodosimetry. Here, we 'knocked out' the microbiome of male and female C57BL/6 mice (Abx mice) using antibiotics and then irradiated (0, 3, or 8 Gy) them to determine the role of the host microbiome on biofluid radiation signatures (1 and 3 d urine, 3 d serum). Biofluid metabolite levels were compared to a sham and irradiated group of mice with a normal microbiome (Abx-con mice). To compare post-irradiation effects in urine, we calculated the Spearman's correlation coefficients of metabolite levels with radiation dose. For selected metabolites of interest, we performed more detailed analyses using linear mixed effect models to determine the effects of radiation dose, time, and microbiome depletion. Serum metabolite levels were compared using an ANOVA. Several metabolites were affected after antibiotic administration in the tryptophan and amino acid pathways, sterol hormone, xenobiotic and bile acid pathways (urine) and lipid metabolism (serum), with a post-irradiation attenuative effect observed for Abx mice. In urine, dose×time interactions were supported for a defined radiation metabolite panel (carnitine, hexosamine-valine-iso-leucine [Hex-V-I], creatine, citric acid, and Nε,Nε,Nε-trimethyllysine [TML]) and dose for N1-acetylspermidine, which also provided excellent (AUROC ≥ 0.90) to good (AUROC ≥ 0.80) sensitivity and specificity according to the area under the receiver operator characteristic curve (AUROC) analysis. In serum, a panel consisting of carnitine, citric acid, lysophosphatidylcholine (LysoPC) (14:0), LysoPC (20:3), and LysoPC (22:5) also gave excellent to good sensitivity and specificity for identifying post-irradiated individuals at 3 d. Although the microbiome affected the basal levels and/or post-irradiation levels of these metabolites, their

DRCCMetadata.php?Mode=Study&StudyID=
ST003000).

**Funding:** This work was funded by a pilot grant (P.
I. ELP) from the Opportunity Funds Management
Core of the Centers for Medical Countermeasures
against Radiation, the National Institutes of Health
(National Institute of Allergy and Infectious
Diseases) grant U19-AI067773 (P.I. David J.
Brenner). The authors acknowledge the Lombardi
Comprehensive Cancer Metabolomics Shared
Resource (MSR), which is in part supported by
Award Number P30CA051008 (P.I. Louis Weiner)
from the National Cancer Institute. The content is
solely the responsibility of the authors and does
not necessarily represent the official views of the
National Cancer Institute or the National Institutes
of Health. The funders had no role in study design,
data collection and analysis, decision to publish, or
preparation of the manuscript. ELP received salary
from the funding agency (National Institute of
Allergy and Infectious Diseases).

**Competing interests:** The authors have declared
that no competing interests exist.

utility in dose reconstruction irrespective of microbiome status is encouraging for the use of metabolomics as a novel biodosimetry assay.

## Introduction

Our current political landscape is witnessing a resurgence of nuclear threats not seen since the Cold War, including a new era of nuclear coercion and a retreat from the New Strategic Arms Reduction Treaty (START) arms-control agreement. In addition, potential malicious actions from terrorist organizations (improvised nuclear devices) and accidents from nuclear power plants (*e.g.*, Chernobyl, Fukushima, Three Mile Island) necessitate that a system of radiation preparedness be available in the event of a large ionizing radiation (IR) exposure to the general population [1]. Within this outline of radiation preparedness includes development of medical countermeasures (MCM), biodosimetry assays, and mitigators/protectors aimed at treating both the acute and long-term effects from IR exposure. For successful biodosimetry assay development, steps have included establishing proper animal models [2], designing novel irradiation systems to mimic complex real-world exposures [3], and using various 'omic' assays for biomarker detection [4]. These goals have been thoroughly outlined within the goals of the Radiation and Nuclear Countermeasures Program within the National Institute of Allergy and Infectious Diseases [5].

Some challenges in biomarker identification for IR exposure includes that pre-exposure samples will not be available for comparison following a radiation emergency. Also, as many individuals will be seeking medical care in a chaotic post-tragedy event, the group may be comprised of varying demographics and genetic makeup. In addition to these factors, the microbiome is known to have profound effects on host metabolite composition in biofluids and tissues that can also be age and sex specific [6, 7]. Microbiota derived metabolites (indole-3-carboxaldehyde and kynurenic acid) have been implicated in host radioprotection [8], although radiation enteritis and intestinal injury can also be lessened in germ free mice or through administering antibiotics [9–11], which is why antibiotics will likely be supplied as a first defense countermeasure in a nuclear emergency [12]. Furthermore, in addition to antibiotic administration [13–16], radiation exposure can have effects on the host microbiome that can affect downstream metabolic analyses [17–19]. From a biodosimetry perspective, it will be necessary to develop a radiation biosignature capable of dose reconstruction irrespective of the natural variation in the human microbiome [20] or antibiotic regimens [21]. Although previous studies have addressed genetic variation on radiation biomarkers [22–24], to date, there have been limited investigations into the role of the host microbiome on small molecule profiles of easily accessible biofluids and how this may affect medical countermeasure efforts in a radiological emergency.

While it is established that IR affects microbiome composition and the metabolome, metabolite presence typically cannot be attributed to a specific organism, such as transcriptomic screening for example, and has hampered our ability to determine specific microbial effects to radiation small molecule biomarkers. The purpose of this study was to determine which biofluid biomarkers may be independent from variations in the host microbiome and provide a stronger case for biodosimetry assays in the event of a radiological emergency. To do this, we provided broad-spectrum antibiotics (enrofloxacin and ampicillin) to male and female C57BL/6 mice to deplete the host microbiome and exposed them to 0, 3, or 8 Gy external IR (Abx mice). We also exposed a treatment group to the above IR doses but did not provide

them with the antibiotic cocktail (Abx-con mice). Both urine (0, 1, and 3 d) and serum (3 d) were collected for small molecule profiling using a liquid chromatography mass spectrometry (LC-MS) platform. As expected, microbiome depletion alone significantly affected several biofluid metabolite levels. To compare post-irradiation effects in urine, we calculated the Spearman's correlation coefficients to identify metabolites that exhibited significant correlations with dose. These select urinary metabolites were then used for more detailed regression analysis using a linear mixed effect model to determine the effects of radiation dose (dose-Gy variable), time (time), and microbiome depletion (microbiome). Serum was collected at 3 d post-irradiation and metabolite levels were compared using an ANOVA. The area under the receiver operating characteristic (AUROC) curves were calculated to determine sensitivity and specificity for both biofluids. We found that the microbiome effects basal levels of many metabolite and post-irradiation levels, with antibiotic administration attenuating the radiation response observed by multivariate analysis. Although microbiome effects were observed, we found that small molecule panels previously identified could differentiated IR exposed individuals irrespective of microbiome status. This is the first study to explore how removal of the host microbiome may affect biofluid metabolites and how this may be a confounder in metabolomics-based dose reconstruction.

## Materials and methods

### Animal experiment and biofluid collection

All animal studies were approved by the Georgetown University Institutional Animal Care and Use Committee (IACUC approved protocol # 2016–1152) and were conducted in facilities accredited by the Association for Assessment and Accreditation of Laboratory Animal Care (AAALAC) and followed all relevant federal and state guidelines. Enrofloxacin and ampicillin were pharmaceutical grade and provided by the Georgetown University Division of Comparative Medicine.

We used 8 week old male and female C57BL/6 mice that were purchased from Charles River Laboratories (Frederick, MD) and were assigned to sham, 3 Gy, or 8 Gy cohorts. The doses and times chosen represent priority groups after a nuclear emergency, as the dose equivalent in humans will be zero damage (sham), minor damage (3 Gy), and major damage (8 Gy, an ~$LD_{50/30}$ value for C57Bl/6 mice). An optimistic expectation would to have all individuals screened within 3 d, however, following the panic ensuing a nuclear emergency we realize that this may take longer. A control group (Abx-con) did not receive antibiotics in drinking water and a treatment group (Abx) received broad-spectrum antibiotics (n = 10 per group, half male half female) (S1 Fig). Mice were provided with deionized water *ad libitum* either with or without the antibiotic cocktail (enrofloxacin [0.575 mg/ml] and ampicillin [1 mg/ml]) for 8 days prior to irradiation [25]. Food (PicoLab Rodent Diet 20 #5053) was provided *ad libitum*. Mice were irradiated in an acrylic, 12-slot mouse pie cage (MPC-1, Braintree Scientific, Braintree, MA) on top of a specimen turntable (XD1905-0000, Precision X-Ray Inc, Branford, CT) and 0, 3 or 8 Gy X-ray irradiated (1.67 Gy/min; X-Rad 320, Precision X-Ray Inc.; filter, 0.75 mm tin/ 0.25 mm copper/1.5 mm aluminum). For Abx-con mice, spot urines (>100 μL) were collected 1 d prior to irradiation and at 1 and 3 d post-irradiation (fecal samples collected 1 d prior to irradiation and at 3 d post-irradiation). For Abx mice, spot urines (>100 μL) were collected 1 d prior to antibiotic administration, 1 d prior to irradiation, and at 1 and 3 d post-irradiation (fecal samples collected 1 d prior to antibiotic administration, 1 d prior to irradiation, and at 3 d post-irradiation). Fecal samples were collected with sterilized tweezers. To aid in dehydration possibly due to ampicillin, which was observed as large abnormal wet stools consistent with low-grade diarrhea [26], we provided the Abx cohort with warm sterilized saline

by hand 1 d prior and during urine collections. At least every 2 d, autoclaved cages, bedding, and water bottles were changed and fresh water with antibiotics and food were provided. Animals were weighed during this time as well as the water bottles to estimate water consumption. The Abx cohort were housed in static cages kept on the top shelf and entry was restricted to 2 researchers on this study and 1 staff member of the Georgetown University Division of Comparative Medicine. Urine and fecal samples were placed on dry ice then immediately stored at −80°C. On day 3, mice were euthanized by $CO_2$ inhalation, blood was collected by cardiac puncture, and tissues were collected. Tissues were flash frozen. Serum samples were prepared using BD Microtainer Tube (REF 365967) with ~100 μL of whole blood added to each tube, kept at room temperature for 30 min, then spun at 1300× g at 4°C for 10 min. All tissues and biofluids were stored at −80°C until analysis.

## DNA extraction and real-time qPCR

DNA extraction and quantitative real-time qPCR targeting the hypervariable region 3 of the 16S ribosomal RNA (HV3-16S) gene was conducted using feces collected 1 d prior to antibiotic administration, 1 d prior to irradiation, and 3 d post-irradiation to determine successful depletion of the microbiome. Plasmids that contained a HV3-16S insert were generously provided by the Blankenship-Paris laboratory (David Goulding, Dr. Blankenship-Paris laboratory, National Institute of Environmental Health Sciences). The plasmid carrying a HV3-16S insert using a TOPO TA Cloning Kit (Invitrogen, Thermo Fisher Scientific, Waltham, MA) and gene inserts were obtained using genomic DNA from *E. coli* strain ATCC 10536. Plasmids were used for standard curve generation and to calculate the number of copies of bacteria present in the feces. Total genomic DNA was extracted from feces using the Qiagen QIAmp PowerFecal DNA Kit (Hilden, Germany) and the DNA quantity was determined using the NanoDrop ND-1000 spectrophotometer (Nanodrop Technologies, Wilmington, DE). DNA samples were diluted to a final concentration of 1 ng/μL using UltraPure DNase/RNase-Free Distilled Water (Invitrogen, Thermo Fisher Scientific, Waltham, MA). Real-time qPCR was performed on each sample in triplicate using 3 μL (3 ng of total DNA template) in combination with SsoAdvanced Universal Inhibitor-Tolerant SYBR Green Supermix (Bio-Rad Laboratories, Hercules, CA) following the manufacture's recommended guidelines. The forward HV3-16S primer 5′CCAGACTCCTACGGGAGGCAG−3′ and the reverse HV3-16S primer 5′−CGTATTACCG CGGCTGCTG−3′(10 μM) were added to the supermix. Real-time qPCR was carried out using a Bio-Rad CFX96 thermal cycler (Bio-Rad, Hercules, CA). Reaction mixtures (20 μL total volume) were held at 98°C for 3 min, followed by 40 cycles at 98°C for 15 s and 60°C for 30 s followed by a melting curve to verify nonspecific amplification. Results are expressed as mean ± std dev.

## Chemicals

For sample extraction and LC analysis we used Fisher Optima™ grade solvents (Fisher Scientific, Hanover Park, IL). The chemical standards were purchased from Cayman Chemical Company (stachydrine hydrochloride [*i.e.*, proline betaine]) (Ann Arbor, MI) or Sigma-Aldrich (debrisoquine, chlorpropamide, 4-nitrobenzoic acid, Nε,Nε,Nε-trimethyllysine hydrochloride [TML], L-carnitine, *cis*-aconitic acid, citric acid, N1-acetylspermidine dihydrochloride, N-acetyl-L-arginine, creatine, niacinamide, 3-methyl-L-histidine, betaine hydrochloride, L-pipecolic acid, indole-3-acetic acid, and indoxyl sulfate) (St. Louis, MO). The hexosamine-valine-isoleucine-OH (Hex-V-I) was synthesized by Expert Synthesis Solutions (London, ON, Canada) with details previously provided [27]. Quality control reference materials were obtained from the National Institute of Standards and Technology (NIST) (NIST

Standard Reference Material 3667 [creatinine in frozen human urine] and NIST Standard Reference Material 1950 [metabolites in frozen human plasma]).

## Biofluid metabolomics

Urine samples were aliquoted (20 μl) and deproteinized using 50% cold acetonitrile (80 μl) with internal standards (2 μM debrisoquine $[M+H]^+$ = 176.1188; 30 μM 4-nitrobenzoic acid $[M-H]^-$ = 166.0141; 5 μM chlorpropamide $[M-H]^-$ = 275.0257). The samples were vortexed, incubated on ice for 10 min, and then centrifuged for 10 min (max speed, 4˚C). A 1 μl aliquot of each urine sample was combined before pre-extraction for a quality control sample and prepared as above. Additionally, the NIST Standard Reference Material 3667 (creatinine in frozen human urine) was used as a quality control sample and prepared as above. The supernatant for each sample was transferred to a sample vial for analysis. A 2 μl aliquot was injected for analysis.

Serum samples were aliquoted (5 μl) and deproteinized using 66% cold acetonitrile (80 μl) with internal standards (2 μM debrisoquine $[M+H]^+$ = 176.1188; 30 μM 4-nitrobenzoic acid $[M-H]^-$ = 166.0141; 5 μM chlorpropamide $[M-H]^-$ = 275.0257). The samples were vortexed, incubated on ice for 10 min, and then centrifuged for 10 min (max speed, 4˚C). A 1 μl aliquot of each serum sample was combined before pre-extraction for a quality control sample and prepared as above. Additionally, the NIST Standard Reference Material 1950 (metabolites in frozen human plasma) was used as a quality control sample and prepared as above. The supernatant for each sample was transferred to a sample vial for analysis. A 2 μl aliquot was injected for analysis.

We used a Waters Acquity Ultra Performance Liquid Chromatography (UPLC) coupled with a BEH C18 1.7 μm, 2.1 x 50 mm column in tandem to a Xevo® G2-S quadrupole time-of-flight (QTOF) MS (Waters, Milford, MA) for analysis. Both positive and negative electrospray ionization (ESI) were used for data acquisition using data-independent acquisition (or $MS^E$). Leucine enkephalin ($[M+H]^+$ = 556.2771, $[M-H]^-$ = 554.2615) was used as the Lock-Spray® reference. Blanks and QC samples were run at the beginning of the run, after every 10 samples, and at the end of the run.

The LC and MS conditions for urine was as follows: LC solvent A (water/0.1% formic acid [FA]) and solvent B (acetonitrile/0.1% FA). Operating conditions for ESI were, capillary voltage 3 kV, cone voltage 30 V, desolvation temperature 500˚C, desolvation gas flow 1000 L/Hr. The gradient for urine was: 4 min 95% A 5% B, 4 min 80% A 20% B, 1.1 min 5% A 95% B, and 1.9 min 95% A 5% B at a flow rate of 0.5 ml/min, column temp 40˚C (S2 Fig).

The LC and MS conditions for serum was as follows: LC solvent A (water/0.1% formic acid [FA]), solvent B (acetonitrile/0.1% FA), and solvent C (isopropanol/0.1% FA). Operating conditions for ESI were, capillary voltage 3 kV, cone voltage 30 V, desolvation temperature 500˚C, desolvation gas flow 600 L/Hr. The gradient for serum was: 4 min 98% A 2% B, 4 min 40% A 60% B, 1.5 min 2% A 98% B, 2 min 11.8% B 88.2% C, 0.5 min 50% A 50% B, and 1 min 98% A 2% B at a flow rate of 0.5 ml/min, column temp 60˚C (S3 Fig).

## Data processing, marker validation, and metabolite modelling

Data pre-processing was performed using MassLynx v.4.1 (Waters Corporation, Milford, MA) and Progenesis QI (Nonlinear Dynamics, Newcastle, UK) for peak alignment using a software chosen QC chromatogram, peak picking, and normalization (all compounds function). Putative identifications (precursor ion ±8 ppm, fragment ion ±20 ppm) were determined by comparing spectral features to the METLIN MS/MS empirical library [28] and the human metabolome database (HMDB) [29]. We mined the post-processed data matrix using

MetaboLyzer [30] using a Welch's *t*-test for features present at ≥70% in both groups or a Barnard's test for features present <70% using a false discovery rate corrected (Benjamini–Hochberg step-up correction procedure) P value of <0.10 in urine or P value <0.20 in serum. To determine the effect of broad-spectrum antibiotics on the urine metabolome without radiation, we combined pre-irradiation and sham groups for Abx-con and Abx mice and generated PCA plots, heatmaps, and volcano plots on the log transformed and pareto scaled post-processed data matrix in MetaboAnalyst 5.0 [31, 32]. Metabolites were plotted in GraphPad Prism 9.2.0 and outliers were detected at ROUT Q = 1% and compared with an unpaired Welch's *t*-test (GraphPad Software, La Jolla, CA). The volcano plots and MetaboLyzer output were used to export excel files for pathway and enrichment analysis using in-house software (Kyoto encyclopedia of genes and genomes [KEGG]), MetaboAnalyst 5.0 (the small molecule pathway database [SMPD]), and MetOrigin [33]. For serum, we compared 3 d post-irradiation samples as above and compared using a Brown-Forsythe ANOVA for post-irradiation effects in Graph-Pad Prism 9.2.0 with outlier detection set at ROUT Q = 1%.

Spectral features of interest were validated to a metabolomics standards initiative (MSI) level 1 by matching the accurate *m/z*, retention time, and tandem MS (5–50 V ramping collision energy) fragmentation patterns to pure standards and the NIST/EPA/NIH Mass Spectral Library 20 v.2.4. For spectral features of interest without a database match (*m/z* 307.2025_5.69, 347.1227_0.27), we used Sirius v5.6.3 to generate the putative elemental formula with high-resolution isotope pattern analysis and the *in silico* fragmentation tree computation to identify the top MS/MS fragments [34].

For post-irradiation effect in urine, modeling of the effects of radiation dose (dose-Gy variable), time (time), and microbiome depletion (microbiome) on the levels of each selected biomarker was performed using *R* 4.2.0 software. To examine relationships between the variables of interest in the data set, we calculated a matrix of Spearman's correlation coefficients and corresponding P values (with Bonferroni correction). We identified metabolites that exhibited significant correlations with dose_Gy, and selected for more detailed regression analysis with time and drug, as follows.

We used the Shapiro-Wilk normality test, QQ-plot, skewness and kurtosis metrics to evaluate how close the distribution of each biomarker of interest was to the normal distribution. If it was not consistent, natural log transformation was used (zero values were replaced with 0.01 constants to allow log transformation). We used multimodel inference (MMI) techniques and linear regression models to find which variables were the strongest and most important predictors of the transformed levels of each biomarker. MMI was implemented using the *glmulti R* package, which calculates Akaike information criterion with sample size correction (AICc) scores for different linear regression models (*lm* function in *R*), ranks the models based on these scores, and estimates normalized relative importance scores and 95% confidence intervals (CIs), corrected for model selection uncertainty, for each predictor. The set of linear regression models was generated by considering all possible combinations of model structures that contained the following variables: dose_Gy, dose_Gy$^2$, time, time$^2$, and microbiome, and multiplicative pairwise interactions between these variables. Those variables and variable interactions which had 95% CIs not overlapping zero, and/or high importance scores, were retained for further analysis, and the others were discarded. Potential multicollinearity between predictor variables was assessed using variance inflation factors (VIF).

A linear model containing the retained variables was fitted, and compared with a robust linear regression model (*lmrob* function in *R*) to assess potential outlier effects on the estimated parameters. Once the preferred model structure was established using the retained predictors, random intercepts and/or slopes which represent inter-individual variability in the responses were considered and fitted using the *lmer* function in *R* (*lme4* package). Mixed effects models

with the same fixed effects and different random effects structures were constructed in this manner, and compared to each other using Akaike information criterion scores. Those model variants which did not converge properly (*e.g.*, due to the random effects structure being too complex for a given data set) were removed from the analysis. The best-supported mixed effects model was assessed using the following diagnostics: visualization and normality testing of residuals using Shapiro-Wilk test, QQ-plot, skewness and kurtosis; and leverage and Cook's distance calculations to detect potentially influential data points. ROC curves were constructed in MetaboAnalyst 5.0 with a Linear SVM classification method for combined metabolites, to determine AUROC values.

## Results

### Microbiome effects on the mouse biofluid metabolome

The mice in this study retained at least 80% of their original body weight and were observed drinking water provided with the enrofloxacin and ampicillin mixture, which indicates problems with palatability experienced with other broad-spectrum antibiotic mixtures were likely insignificant [25]. There was no difference in the amount of water consumed between cages or sexes, where the males consumed an average of 15 g water/day and females consumed 14 g water/day. We did observe cecum enlargement of mice post-mortem as others have previously reported (Fig 1A). There were also sex specific differences in microbiome depletion efficiency, where females had a stronger response to drug treatment than males (S4 Fig). Female mice had significantly lower levels of microbial DNA in the feces after the administration of antibiotics. Male mice had a significantly lower level of microbial DNA before irradiation and after the 8-day antibiotic regimen, with a return to pre-antibiotic levels after irradiation (S4 Fig).

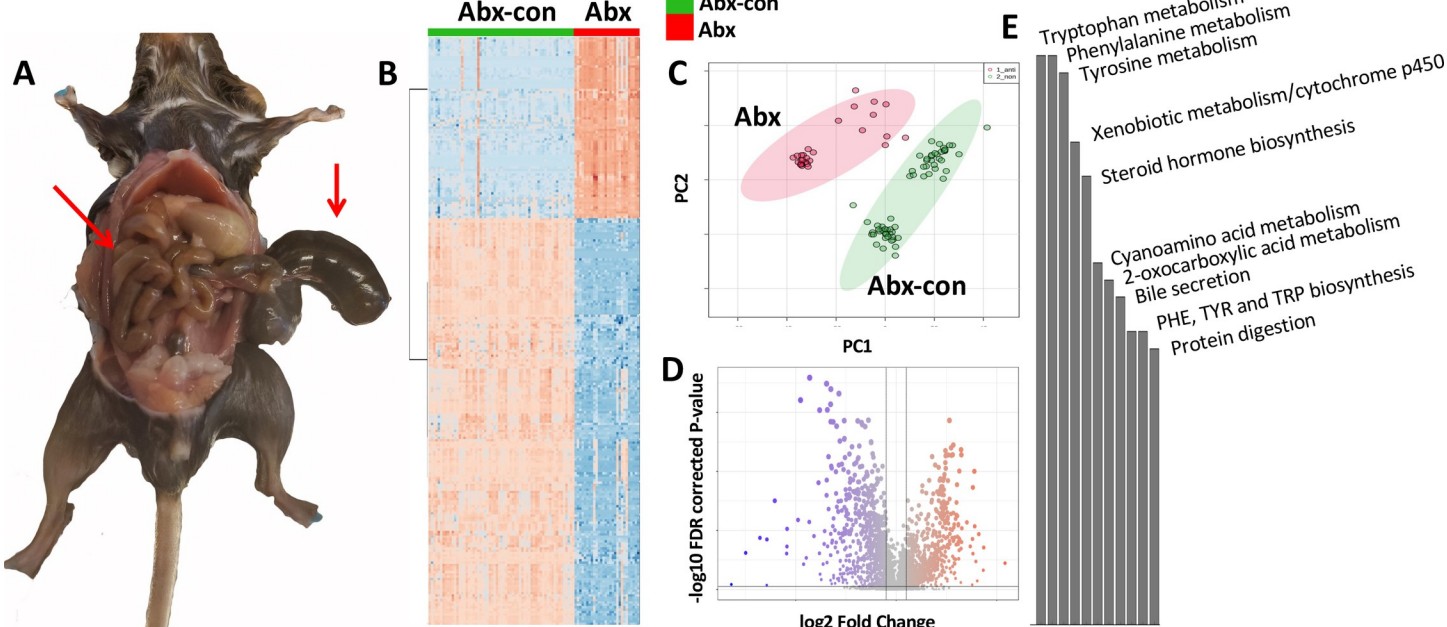

**Fig 1. Contribution of the host microbiota on the mouse urine metabolome: Part 1.** A) Abx mice showed an enlarged cecum compared to Abx-con mice. Large abnormal wet stools were also observed that may be consistent with low-grade diarrhea from ampicillin administration. B-D) Heatmap, PCA plot, and volcano plot of positive mode spectral features showing perturbation to the urinary metabolome following depletion of the gut microbiome. Significant spectral features were extracted from the volcano plot to perform E) pathway analysis with in-house software showing that the major pathways being contributed to by the host microbiome include tryptophan metabolism in addition to amino/cyanoamino acid metabolism/digestion, sterol hormone, xenobiotic, and bile acid metabolism.

Although fecal microbial DNA levels were not significantly lower after irradiation in male mice compared to their pre-antibiotic samples, their urinary metabolome was strikingly similar at all time points after receiving the antibiotic regimen and significantly different from the pre-antibiotic time point. This discrepancy may be due to hyperproliferation of microorganism species not selected for in the broad-spectrum antibiotic regimen, as has been observed in single dose oral gavage delivery methods in previous studies [35].

There are obvious changes in the urine metabolome following host microbiome depletion in both the male and female cohorts (Fig 1B–1D). The volcano plot shows 29.7% of spectral features detected in ESI+ were significantly affected by the host microbiome. Pathway analysis shows that the primary metabolites driving the changes in Abx-con vs. Abx mice were involved in tryptophan metabolism in addition to amino acid, sterol hormone, xenobiotic and bile acid metabolism (Fig 1E). As expected, a majority of the significantly different metabolites between Abx-con and Abx mice had microbial, co-metabolism, drug metabolites, or food related origins (Fig 2). We did observe 24 metabolites of host origin, however, a majority (15) of these were putative steroids or steroid derivatives that can be perturbed by microbiome depletion [36]. Most metabolites involved in indole metabolism or responses to IR exposure were statistically lower after microbiome depletion in the female cohort, excluding Hex-V-I

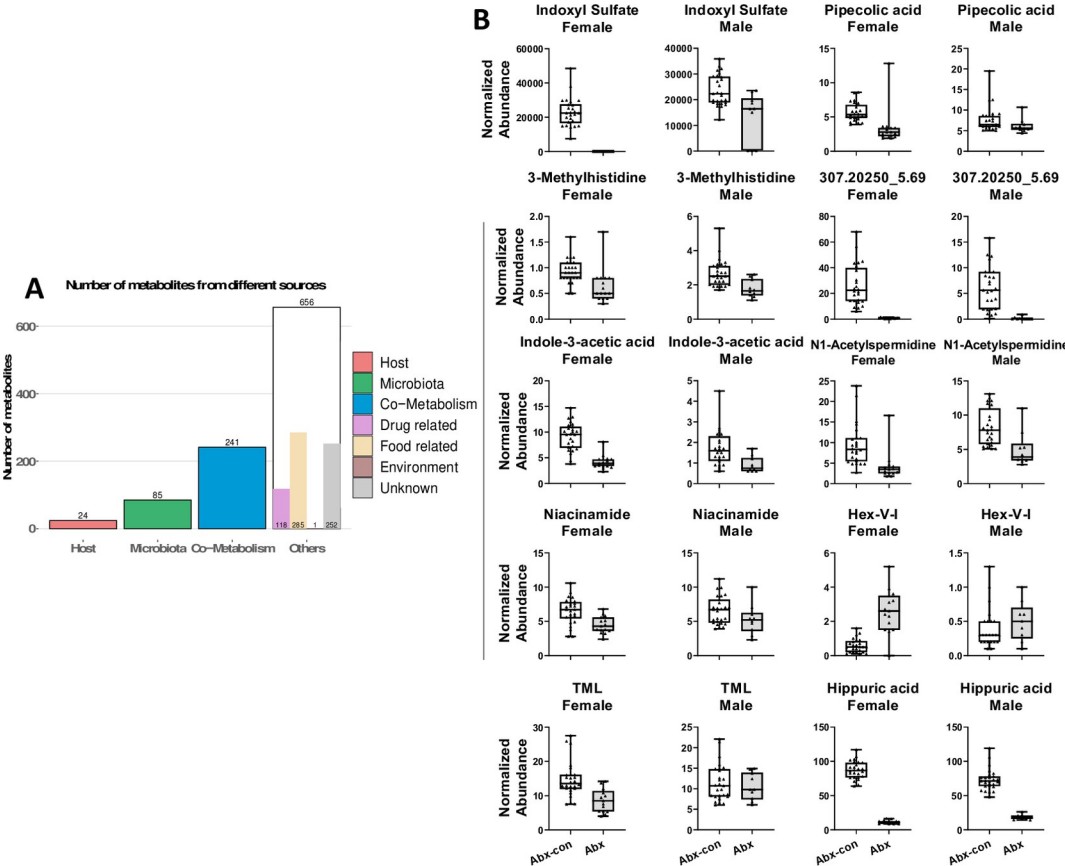

**Fig 2. Contribution of the host microbiota on the mouse urine metabolome: Part 2.** A) Origin of statistically significant metabolites between Abx-con and Abx mice (putative identification), showing that the majority of metabolites are microbial, co-metabolism, drug metabolites, and food related. Although 24 metabolites were identified as host metabolites, 15 of these are steroids or steroid derivatives that may be affected by the host microbiota. B) Several compounds that are typically involved in indole metabolism or ionizing radiation responses were found to be significantly affected by microbiome depletion.

(S1 Table). As the DNA levels unexpectedly remained high in the Abx males despite having a significantly different urinary microbiome, significant differences were still present in these metabolites excluding Hex-V-I, TML, N1-acetylspermidine, and pipecolic acid. Microbiome depletion had fewer effects on the serum metabolome compared to urine, where 10.2% of spectral features, primarily corresponding to lipids, changed at a statistically significant level (S5 Fig).

## Microbiome effects on the mouse biofluid metabolome after ionizing radiation exposure

**Urinary metabolomics.** As reported in several previous studies [37], we see that IR exposure induces consistent and broad changes in the urine metabolome as depicted in a heatmap, PCA plot, and volcano plot for the Abx-con mice [38] (Fig 3A). As a certain percentage of these metabolites are microbially produced [39], depletion of the host microbiota should reduce the total number of significantly changed urinary spectral features observed post-irradiation. Consistent with this reasoning, we observed lower fold changes in the top 250 positive mode ions depicted in heatmaps and less separation in a PCA plot in Abx mice at 1 d following an 8 Gy exposure (Fig 3A Abx-con mice, Fig 3B Abx mice), which is the dose and time in the current study that will show the highest perturbation. Additionally, a higher number of urine features in volcano plots were significant in the control group (312 features, Fig 3A) compared to microflora depleted group (201 features, Fig 3B), including metabolites in amino acid and pyrimidine metabolism, propionic acid metabolism, hormone synthesis, and ammonia recycling (S6 Fig). Perturbation to pathways that were conserved post-irradiation irrespective of the host microbiota included energy metabolism, including fatty acid ß oxidation and the TCA cycle, and lipid metabolism.

Several typical radiation urinary small molecule markers that we have previously identified [37] were significantly perturbed after 3 or 8 Gy IR exposures in both Abx-con and Abx mice

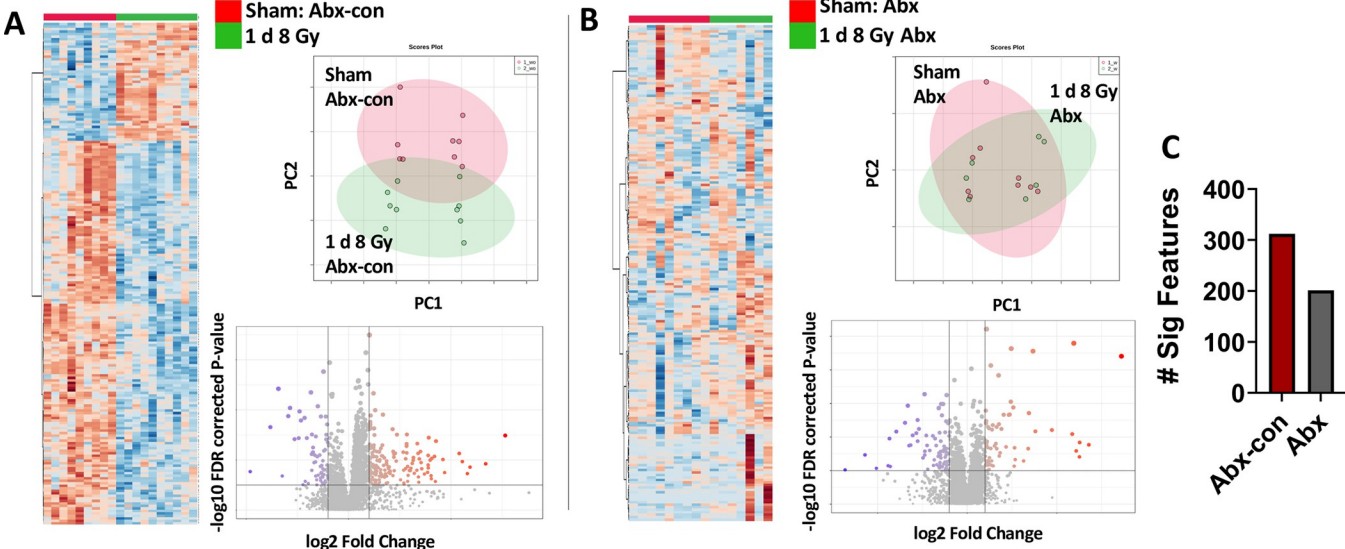

**Fig 3. Influence of the mouse microbiome on the urine metabolome post-irradiation.** Heatmaps (same spectral features for both), PCA plots, and volcano plots showing broad changes in mouse urine at 1 day post-irradiation at a 8 Gy dose in a A) Abx-con mice compared to B) Abx mice. C) Number of significant spectral features between Abx-con and Abx mice. The control group shows typical changes at 1 day post-irradiation at high ionizing radiation doses, where significant perturbation to the urinary metabolome is observed. While several spectral features were conserved in mice given broad spectrum antibiotics, an overall reduction in fold change and number of statistically significant features were observed. Although some of the features that were absent are directly due to lack of microbial metabolism, a possible protective effect may also play a role.

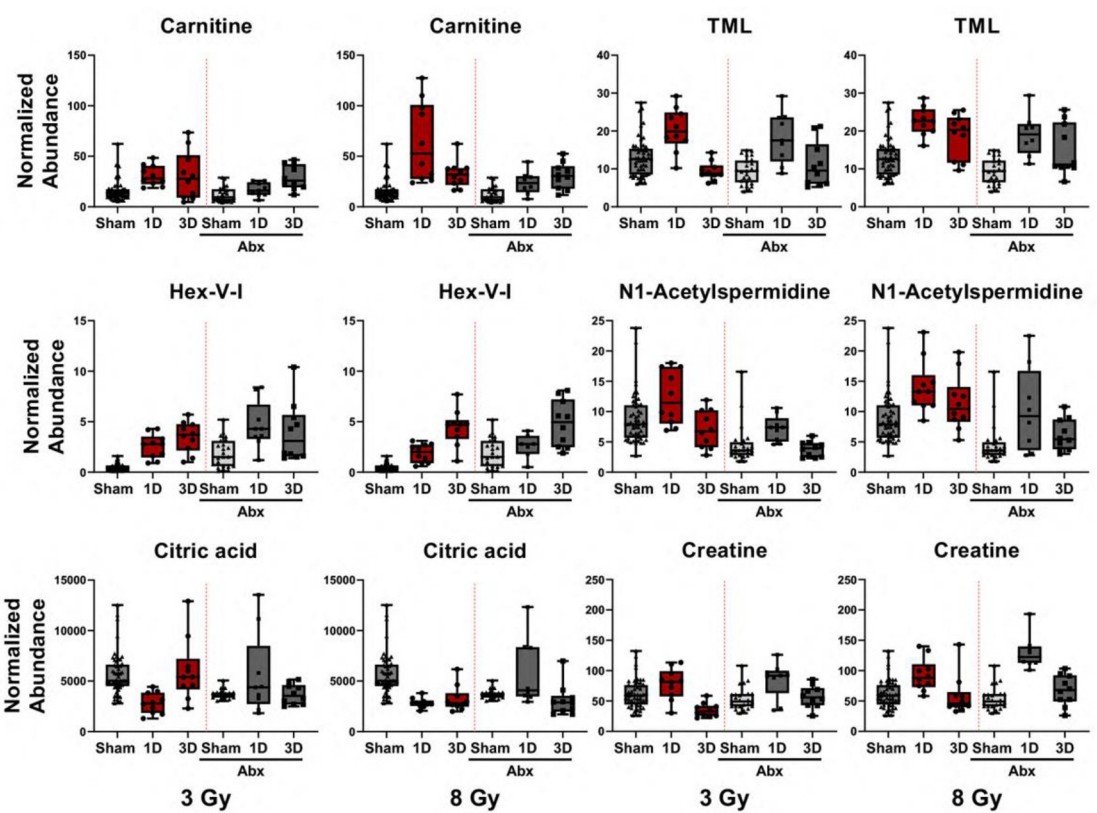

**Fig 4. Validated urinary metabolites after IR exposure.** Concentrations of validated urinary metabolites in Abx-con or Abx mice at 1 and 3 days following a 3 or 8 Gy TBI.

(Fig 4, S7, S8 Figs, Table 1 and S2 Table). We then narrowed down the number of metabolites by determining which ones strongly correlated with dose using Spearman's correlation coefficients, and/or appeared promising to investigate in more detail. These compounds included Hex-V-I, creatine, carnitine, TML, N1-acetylspermidine, and citric acid (Table 2). The combination of biomarkers (also including proteins, genes, etc.) will likely be required due to the natural variation in a human population, for example outliers can be observed even in from control C57Bl/6 mice in the current study (Fig 4). To further elucidate interactions between the host microbiome and IR exposure and additional covariates (dose and time effects), we used a mixed effects modeling approach to account for both fixed and random effects and estimate individual-level effects of these metabolites. We observed both microbiome and dose×time interaction effects for these urinary markers. Quadratic interaction terms involving dose$^2$ and/or time$^2$ were also supported in some cases. Such interactions imply that the effect of one predictor variable (*e.g.*, radiation dose) on the response variable is not constant, but rather varies depending on the value of another predictor variable (*e.g.*, time). A quadratic interaction can be interpreted as follows: Imagine we have two predictor variables, dose and time, and we want to model their joint effect on the response variable, Y (*i.e.*, a biomarker type). A quadratic interaction between dose and time means that the effect of dose on Y depends not only on the value of dose, but also on the value of time.

The microbiome and dose×time interaction variables were retained in the model as predictors of the natural log of four biomarkers (ln_carnitine, ln_Hex-V-I, and ln_creatine). For carnitine (dose×time P value = <0.001, microbiome P value = <0.001), the interaction variables

**Table 1. Validated urinary metabolites.**

| Metabolite | Adduct | RT | Experimental (m/z) | Calculated (m/z) | Mass Error (ppm) | HMDB | Formula | MS/MS Fragments | | |
|---|---|---|---|---|---|---|---|---|---|---|
| | | | | | | | | Fragment 1 | Fragment 2 | Fragment 3 |
| Hex-V-I[#] | H+ | 1.34 | 393.2239 | 393.2234 | 1.3 | 162421477* | $C_{17}H_{32}N_2O_8$ | 309.1853 | 216.1259 | 150.0921 |
| Creatine | H+ | 0.29 | 132.0775 | 132.0773 | 1.5 | 0000064 | $C_4H_9N_3O_2$ | 114.0629 | 90.0553 | 44.0513 |
| Carnitine | H+ | 0.29 | 162.1132 | 162.1130 | 1.2 | 0000062 | $C_7H_{16}NO_3$ | 103.0397 | 85.0289 | 60.0820 |
| TML[#] | H+ | 0.27 | 189.1606 | 189.1603 | 1.6 | 0001325 | $C_9H_{20}N_2O_2$ | 130.0873 | 84.0818 | 60.0809 |
| N1-Acetylspermidine[#] | H+ | 0.27 | 188.1762 | 188.1763 | 0.5 | 0001276 | $C_9H_{21}N_3O$ | 171.1504 | 117.1013 | 100.0766 |
| Citric acid[#] | H- | 0.32 | 191.0193 | 191.0192 | 0.5 | 0000094 | $C_6H_8O_7$ | 173.0088 | 111.0088 | 87.0088 |
| Betaine | H+ | 0.29 | 118.0869 | 118.0868 | 0.8 | 0000043 | $C_5H_{12}NO_2$ | 59.0743 | 58.0641 | - |
| Niacinamide[#] | H+ | 0.30 | 123.0560 | 123.0558 | 1.6 | 0001406 | $C_6H_6N_2O$ | 106.0237 | 96.0455 | 80.0504 |
| Proline Betaine[#] | H+ | 0.31 | 144.1026 | 144.1025 | 0.7 | 0004827 | $C_7H_{13}NO_2$ | 102.0557 | 84.0817 | 72.0823 |
| Pipecolic acid[#] | H+ | 0.30 | 130.0876 | 130.0868 | 6.1 | 0000070 | $C_6H_{11}NO_2$ | 84.0815 | 56.0423 | 55.0523 |
| 3-Methylhistidine[#] | H+ | 0.25 | 170.0931 | 170.0929 | 1.2 | 0000479 | $C_7H_{11}N_3O_2$ | 124.0882 | 109.0770 | 96.0688 |
| Hippuric acid[#] | H+ | 1.60 | 180.0663 | 180.0661 | 1.1 | 0000714 | $C_9H_9NO_3$ | 162.0563 | 134.0608 | 105.0343 |
| Acetyl-arginine | H+ | 0.31 | 217.1302 | 217.1301 | 0.5 | 0004620 | $C_8H_{16}N_4O_3$ | 175.1195 | 158.0812 | 70.0661 |
| 307.2025_5.69**[#] | H+ | 5.69 | 307.2025 | 307.2022 | 1.0 | - | $C_{17}H_{26}N_2O_3$ | 247.1810 | 183.1215 | 164.1075 |
| 347.1227_0.27** | K+ | 0.27 | 347.1227 | 347.1221 | 1.8 | - | $C_{12}H_{24}N_2O_7$ | 268.9492 | 227.0812 | 222.9420 |
| Indole-3-acetic acid[#] | H+ | 3.98 | 176.0713 | 176.0712 | 0.6 | 0000197 | $C_{10}H_9NO_2$ | 130.0654 | 103.0544 | 77.0378 |
| Indoxyl Sulfate[#] | H- | 1.41 | 212.0026 | 212.0018 | 3.8 | 0000682 | $C_8H_7NO_4S$ | 132.0454 | 80.9647 | 79.9569 |
| *cis*-Aconitic acid[#] | H- | 0.34 | 173.0089 | 173.0086 | 1.7 | 0000072 | $C_6H_6O_6$ | 129.0181 | 111.0080 | 85.0290 |

* Pubchem CID

**Putative adduct and elemental formula from top hit in Sirius 5.6.3

[#]Microbial effect

were not improved using random effects, and they suggested an increase in ln_carnitine due to dose×time and an increase with a normal microbiome. Random effects for the intercept were supported for ln_Hex-V-I (dose×time P value = <0.001, microbiome P value = <0.001) and ln_creatine (dose×time P value = <0.001, microbiome P value = 0.006). The dose variable alone was significant for ln_acetylspermidine (dose×time P value = 0.028), however, the dose×time interaction was not significant using random effects but the microbiome interaction was retained (dose×time P value = 0.201, microbiome P value = <0.001). Only the dose×time interaction was retained in the model as predictors of ln_TML (dose×time P value = 0.001) and ln_citric acid (dose×time P value = <0.001). Allowing random effects for the intercept improved both models, based on AIC scores. Mixed effects model output including the regression parameters, standard errors, p-values, and diagnostics on regression residuals are supplied in S3 Table.

**Table 2. P-value and correlations with dose from Spearman's correlation coefficients for urinary metabolites that were significantly perturbed due to ionizing radiation exposure.**

| Metabolite | P-value | Correlation |
|---|---|---|
| Hex-V-I | <0.001 | + |
| Creatine | 0.001 | + |
| Carnitine | <0.001 | + |
| TML | <0.001 | + |
| N1-Acetylspermidine | <0.001 | + |
| Citric acid | 0.001 | - |

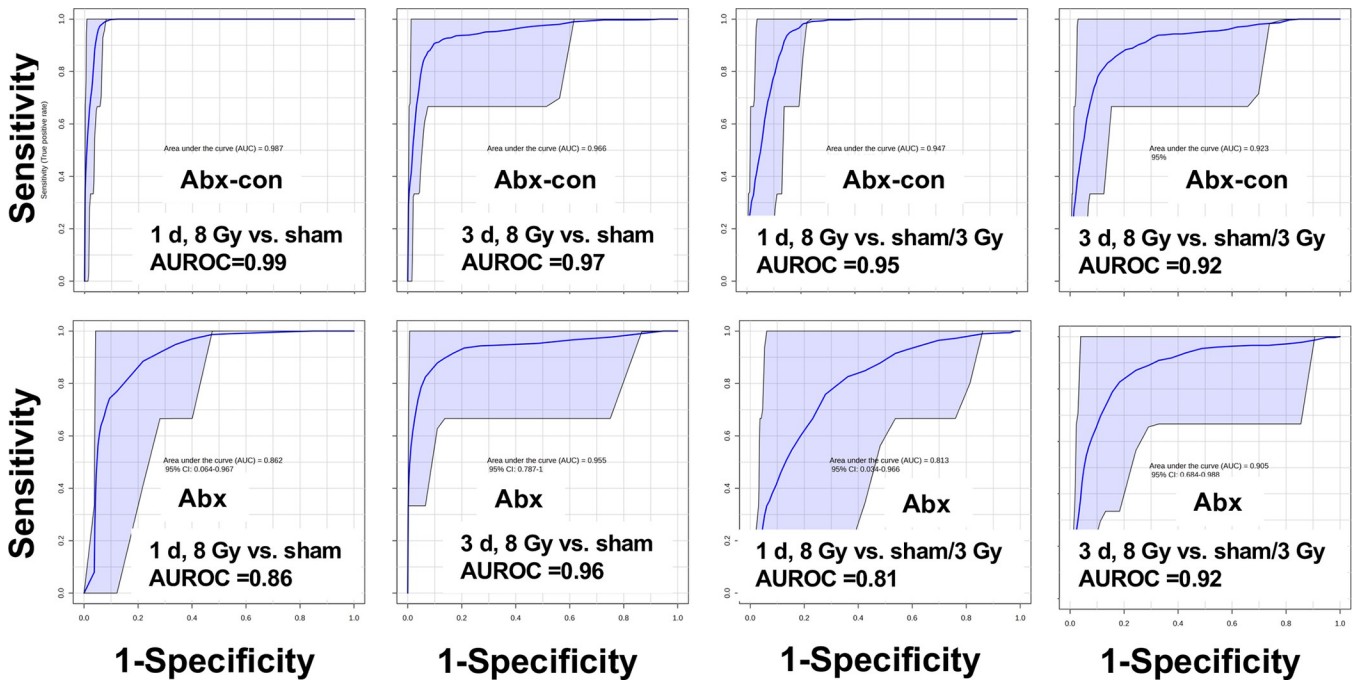

**Fig 5. ROC analysis of urinary metabolites in mice after IR exposure.** The AUROC values show excellent (AUROC ≥ 0.90) to good (AUROC ≥ 0.80) sensitivity and specificity when constructed using five common markers of IR exposure in both Abx-con and Abx mice: including carnitine, citric acid, TML, Hex-V-I, N1-acetylspermidine, and creatine. Creatine levels are perturbed only at 1 d and are not included in 3 d ROC curves.

Other interesting spectral features included one that showed significant changes due to IR exposure in the Abx-con mice but was absent in Abx mice (*m/z* 307.2025_5.29) and another showed similar perturbation to IR exposure in both groups (*m/z* 347.1227_0.27). As the acquired accurate *m/z* and tandem MS spectra (S8 Fig) did not correspond to known compounds in online databases or the NIST library, we used Sirius v5.6.3 to determine putative adducts and molecular formulas (Table 1) and their identity remains an open area of research.

We further assessed the sensitivity and specificity of the metabolite panel with significant dose correlations by examining the AUROC and confidence intervals from ROC curves (Fig 5). While excellent (AUROC ≥ 0.90) to good (AUROC ≥ 0.80) sensitivity and specificity was achieved for both Abx-con and Abx mice, the Abx mice show lower AUROC values and wider confidence intervals compared to Abx-con mice, which indicates that depleting the host microbiome negatively impacted dose reconstruction. Comparing the 8 Gy exposure group to only the sham group, a combination of six biomarkers (carnitine, citric acid, TML, Hex-V-I, N1-acetylspermidine, and creatine) gave nearly perfect classification performance for Abx-con mice (AUROC = 0.99), however, lower fold changes for Hex-V-I and carnitine in the Abx mice gave a lower AUROC (0.86). Similar performance between Abx-con and Abx mice was observed at 3 d using five metabolites (carnitine, citric acid, TML, Hex-V-I, and N1-acetylspermidine), which is still a critical time for evaluating individuals following a radiation emergency. Creatine values were close to basal levels by 3 d at both 3 and 8 Gy exposures. We also compared the 8 Gy cohort against both sham and 3 Gy exposed individuals. Excellent (AUROC ≥ 0.90) to good (AUROC ≥ 0.80) sensitivity and specificity was still achieved for both Abx-con and Abx mice at 1 d using a four-metabolite panel (carnitine, creatine, TML, and N1-acetylspermidine) and at 3 d using the identical five metabolite panel as above (carnitine, citric acid, TML, Hex-V-I, and N1-acetylspermidine).

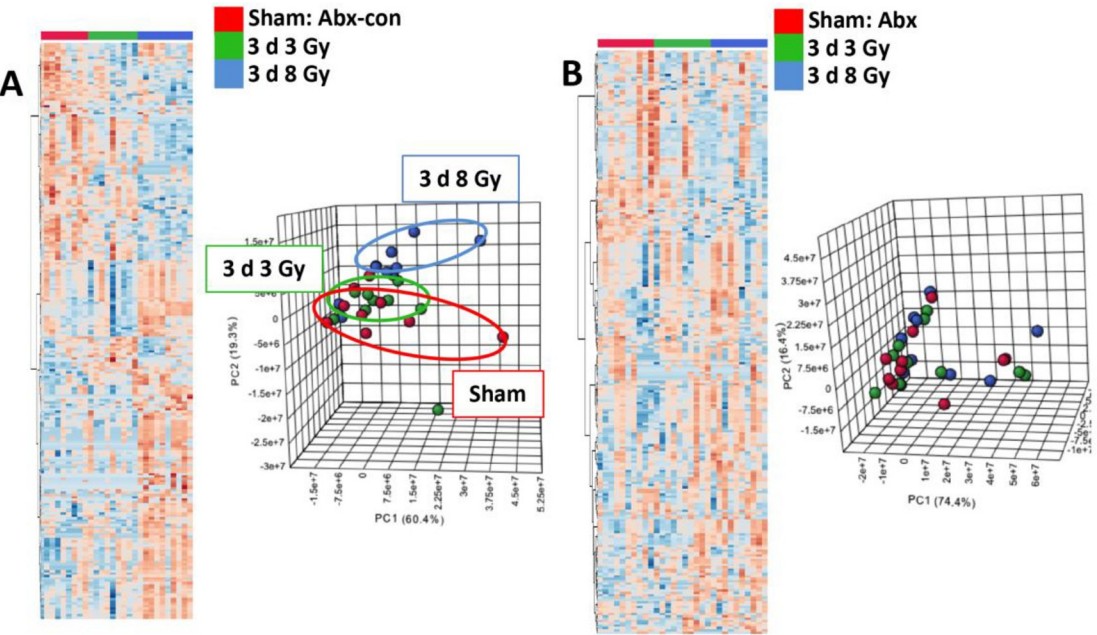

**Fig 6. Influence of the mouse microbiome on the serum metabolome post-irradiation.** Heatmaps and PCA plots showing changes in mouse serum at 3 days post-irradiation at a 3 or 8 Gy dose in a A) control group (no antibiotics) compared to B) mice with a depleted microbiota. Similar to urine, we see higher fold changes in the top 250 spectral features in ESI+ mode and better separation in a PCA plot.

**Serum metabolomics.** Serum was collected via cardiac puncture, so 3 d samples were compared for the sham group to the 3 or 8 Gy IR exposure groups. Similar to urine, we observed more defined changes due to IR exposure in mice with a normal microbiota compared to mice with a depleted microbiota (Fig 6). Although sham irradiated mice in the Abx-con group show a clear separation from mice exposed to 8 Gy IR, feature fold changes were more diffuse in the Abx irradiated groups and little separation was observed for the top three principal components.

As with urine, similar trends were observed post-irradiation between Abx-con and Abx mice for common radiation metabolites that have been previously identified [37, 40] (Fig 7). A statistically significant decrease was observed in proline betaine after an 8 Gy (Abx-con FC = 0.6; Abx FC = 0.6) exposure for both Abx-con and Abx mice and in citric acid after both 3 (Abx-con FC = 0.5; Abx FC = 0.6) and 8 Gy (Abx-con FC = 0.4; Abx FC = 0.4) exposures (Table 3, S4 Table). Lower fold changes were observed for serum carnitine levels in the current experiment compared to past studies, although its inclusion increased the sensitivity and specificity of ROC curves.

Several interesting lysophosphatidylcholines (LysoPCs) showed decreases for both Abx-con and Abx mice, however, statistically significant changes were primarily observed at 8 Gy. At 3 Gy, only LysoPC (14:0) (Abx-con FC = 0.6) and LysoPC (20:3) (Abx FC = 0.6) were statistically significant and increased ROC curve sensitivity and specificity. After an 8 Gy exposure, similar decreases were observed in LysoPC (14:0) (Abx-con FC = 0.5; Abx FC = 0.6), LysoPC (20:3) (Abx-con FC = 0.5; Abx FC = 0.5), and LysoPC (20:5) (Abx-con FC = 0.6; Abx FC = 0.6) for both Abx-con and Abx mice, while decreases in LysoPC (18:3) (Abx-con FC = 0.7) was only statistically significant in Abx-con mice and LysoPC (16:1) (Abx FC = 0.6) and LysoPC (22:5) (Abx FC = 0.5) for Abx mice.

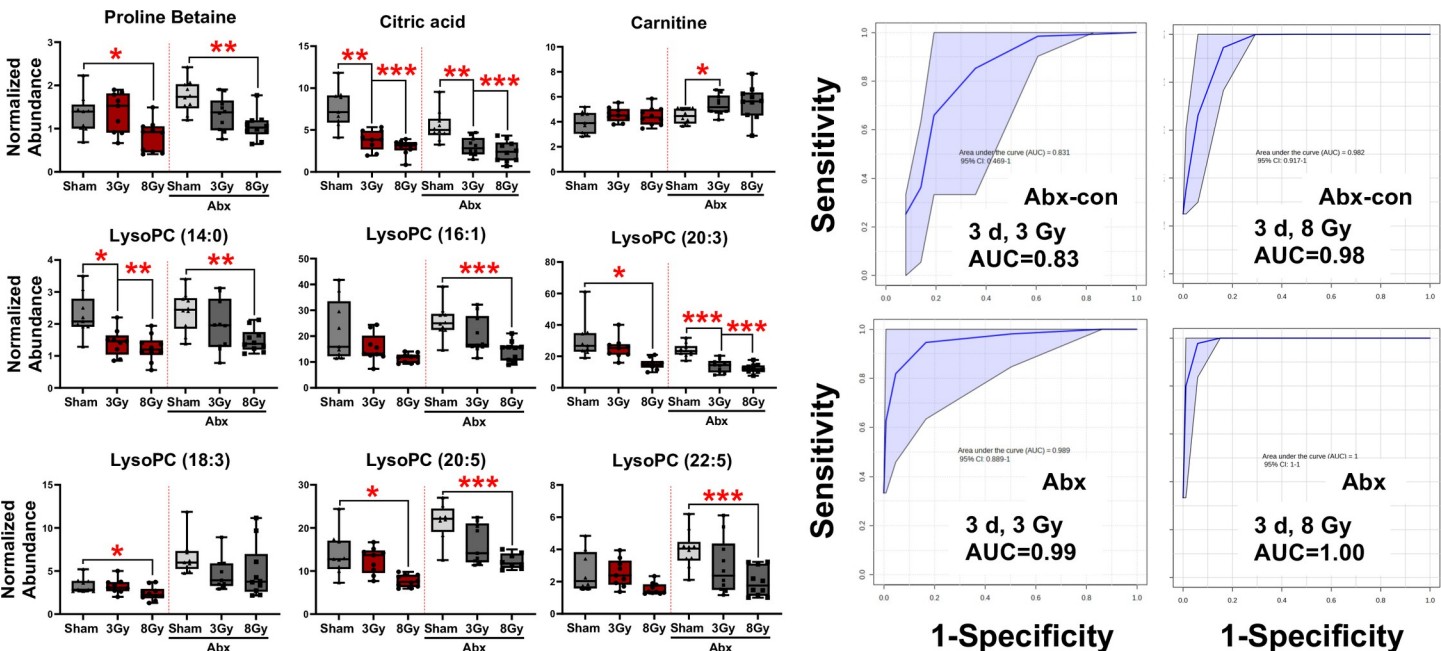

**Fig 7. Validated serum metabolites and ROC analysis after IR exposure.** Concentrations of validated serum metabolites in Abx-con or Abx mice at 3 days following a 3 or 8 Gy TBI. The AUROC values show excellent (AUROC ≥ 0.90) to good (AUROC ≥ 0.80) sensitivity and specificity is achieved using carnitine, citric acid, LysoPC (14:0), LysoPC (20:3), and LysoPC (22:5).

ROC curve analysis showed that excellent to good sensitivity and specificity is achieved comparing the 3 Gy cohort to the sham cohort using a panel comprised of carnitine, citric acid, LysoPC (14:0), LysoPC (20:3), and LysoPC (20:5). Excellent sensitivity and specificity and smaller confidence intervals were observed for the identical panel comparing the 8 Gy cohort to the sham cohort.

## Discussion

In our previous studies, we addressed how genetic variation in the general population may affect small molecule biofluid signature post-irradiation through radiosensitivity [23, 40], radioresistance [24], or immunosuppressive conditions [41]. In addition to host genetic variation, our individual microbiome composition varies greatly not only between people, but can

**Table 3. Validated serum metabolites.**

| Metabolite | Adduct | RT | Experimental (m/z) | Calculated (m/z) | Mass Error (ppm) | HMDB | Formula | MS/MS Fragments | | |
|---|---|---|---|---|---|---|---|---|---|---|
| | | | | | | | | Fragment 1 | Fragment 2 | Fragment 3 |
| Proline Betaine | H+ | 0.29 | 144.1026 | 144.1025 | 0.7 | 0012273 | $C_{16}H_{33}NO$ | 102.0529 | 84.0816 | 72.0847 |
| Carnitine | H+ | 0.29 | 162.1128 | 162.1130 | 1.2 | 0000062 | $C_7H_{16}NO_3$ | 103.0395 | 85.0285 | 60.0804 |
| Citric acid | H- | 0.35 | 191.0187 | 191.0192 | 2.3 | 0000094 | $C_6H_8O_7$ | 173.0033 | 111.0099 | 87.0091 |
| LysoPC (14:0) | H+ | 4.75 | 468.3089 | 468.3090 | 0.2 | 0010379 | $C_{22}H_{46}NO_7P$ | 450.2954 | 184.0734 | 104.1075 |
| LysoPC (16:1) | H+ | 4.88 | 494.3245 | 494.3247 | 0.4 | - | $C_{24}H_{48}NO_7P$ | 476.3133 | 184.0741 | 104.1078 |
| LysoPC (18:3) | H+ | 4.83 | 518.3247 | 518.3247 | 0.0 | - | $C_{26}H_{48}NO_7P$ | 500.3125 | 184.0736 | 104.1073 |
| LysoPC (20:5) | H+ | 4.81 | 542.3239 | 542.3247 | 1.5 | - | $C_{28}H_{48}NO_7P$ | 524.3122 | 184.0739 | 104.1079 |
| LysoPC (20:3) | H+ | 5.26 | 546.3561 | 546.3560 | 0.2 | - | $C_{28}H_{52}NO_7P$ | 528.3445 | 184.0735 | 104.1076 |
| LysoPC (22:5) | H+ | 5.18 | 570.3564 | 570.3560 | 0.7 | - | $C_{30}H_{52}NO_7P$ | 552.3449 | 184.0737 | 104.1077 |

change due to dietary habits and age. The host microbiome is integral to our overall health and impacts metabolite composition in all tissues and biofluids [6, 16]. Also, global consumption of antibiotics since 2000 has risen >40% [21] and are recommended as a therapeutic in radiation emergencies to aid in GI syndrome [10, 12]. Given the effects this may have on dose reconstruction in biodosimetry, we used a broad-spectrum antibiotic regimen to deplete the microbiome in a murine model and determine the interactions between the microbiome, IR doses that cause minimal (3 Gy) to severe (8 Gy, our $LD_{50/30}$ dose) radiation injury, and time that individuals would be seeking medical treatment following a radiation emergency (1 and/ or 3 d). As others have previously reported [13, 42], we found that the microbiome has a profound effect on the biofluid metabolome. However, several common markers of IR exposure are perturbed irrespective of their microbiome status and can be used for dose reconstruction with high sensitivity and specificity, further strengthening the case for small molecule biodosimetry.

Several studies have been dedicated to determining the role of the host microbiome in biofluid metabolite composition. Mice provided with a broad-spectrum β-lactam antibiotic, imipenem, over 4 days showed perturbation in urinary metabolites involved in amino acid metabolism, TCA cycle, and purine and pyrimidine metabolism, however these levels recovered after the mice stopped receiving the antibiotic [13]. Humans receiving β-lactam antibiotic alone or in addition to other medications showed correlating changes in blood metabolites (*e. g.*, hippuric acid, indole propionate, indoxyl sulfate, cresol sulfate) as had been recorded in murine models [15]. Hippuric acid and indoxyl sulfate were two top metabolites associated with the host microbiota in urine and serum in germ free mice [6] and depleted in the urine of mice given a cocktail of neomycin, streptomycin, and bacitracin [43]. Plasma levels of hippuric acid, indole-3-acetic acid, amino acids, and glycerol in rats were perturbed across several antibiotic classes while changes in indoxyl sulfate was associated with tetracyclines [42], however, we found in the current study urinary levels of indoxyl sulfate are heavily affected by a combination of a fluoroquinolone (enrofloxacin) and ampicillin as well. While sex differences are observed in the normal gut microbiota [44], differential response to antibiotics have also been observed in Sprague Dawley rats given a mixture of vancomycin, ampicillin, neomycin, and metronidazole [45]. Male rats showed an increased loss of microbial diversity than females in the stool, the relative abundance of *Lactobacillus*, *Akkermansia*, *Serratia*, and *Sutterrella* was higher in male stool [45]. Although changes in the microbiome can be treatment and sex specific, we find several similarities across previous studies using this broad-spectrum antibiotic mixture, which induced changes in several metabolic pathways including tryptophan and amino acid metabolism, bile secretion, and steroid hormone biosynthesis.

In addition to effects attributed to the microbiome alone, there is an interplay between the microbiome and recovery from IR injury. Recently, several studies have been dedicated to this subject spanning low dose rate environmental studies [46], radiotherapy [47], and medical countermeasures [48, 49]. Given the wealth of information on this topic along with fairly recent reviews [11, 48], a comprehensive review here is beyond the scope of the study. What has been less explored, is the interplay between the microbiome, IR injury, and small molecule biomarkers for dose reconstruction beyond observational correlation to known microbial pathways (*e.g.*, tryptophan metabolism) [50]. Our group published an earlier study on this topic, which used an inter-omic correlation analysis to determine associations between the fecal metabolome and changes in the host microbiome in a murine model following a 5 Gy (3, 14, and 30 d) or 12 Gy (3 d) TBI exposure [17]. While the correlation analysis associated the highest with *Firmicutes* and *Bacteroidetes*, the primary metabolite pathway perturbation was observed in tryptophan, tyrosine, cyanoamino acid, and phenylalanine metabolism and bile acid secretion. Others correlated the microbiome to negative metabolic health from high fat

diet and low dose radiation co-exposure [18] and showed that "elite-survivors" of IR doses (8–9.2 Gy) maintain higher *Lachnospiraceae* and *Enterococcaceae* levels and that an increase in microbial short chain fatty acids and tryptophan metabolites may attenuate GI damage [8].

Changes in urinary metabolite levels post-irradiation have been documented in several animal models and patient samples spanning different dose rates and times [37, 38, 51]. A common theme present in many of these studies has been that energy metabolism is perturbed (*e. g.*, TCA cycle intermediates and fatty acid ß oxidation) after exposure to IR and that metabolites in these pathways may be reliable for biodosimetry. In fact, the effects of IR exposure on the TCA cycle and mitochondrial metabolism have been documented for several decades [52, 53]. We find reduced levels of most TCA cycle intermediates in urine in NHP models if utilizing GC platforms [54, 55], however, citric acid is most often reported due to the ubiquity of LC platforms being used for these descriptions (or NMR [56]). Pre-irradiation, we found significant reductions in both citric acid and *cis*-aconitic acid from microbiome depletion. Lowered levels of urinary TCA cycle intermediates from microbiota depletion are interesting as the host microbiome has been implicated in increasing circulatory levels of succinate [57]. Post-irradiation, we found typical decreases at 1 d for both of these metabolites after 3 and 8 Gy exposures in Abx-con mice, but no statistically significant changes were seen in Abx mice. The lower fold changes in Abx mice may be partly responsible for the increased confidence intervals observed in the ROC analysis and could negatively affect dose reconstruction. Changes in fatty acid ß oxidation precursors and intermediates (carnitine, TML, acetylcarnitine, and acylcarnitines) post-irradiation has been recently reviewed [58]. Pre-irradiation, there is a microbial effect on the basal urinary levels of TML, but not free carnitine and acylcarnitine levels. Post-irradiation, mixed model interaction variables show significant microbiome effects on carnitine only, with TML being only significant for dose×time interactions. Irrespective of the microbiome effects, carnitine, TML, and citric acid were retained in all ROC curve analyses, although citric acid did lower the predictive performance for Abx mice.

Other radiation markers previously identified include Hex-V-I, N1-acetylspermidine, and creatine, of which Hex-V-I and N1-acetylspermidine show significantly different levels in pre-irradiation samples between Abx-con and Abx mice. Post-irradiation mixed model interactions were significant for the microbiome and dose×time for all three markers as well. While creatine perturbation has been well documented in several previous studies at early time points (1 d) [38], we have found it quickly returns to basal levels by day 3 which limits its temporal effectiveness [59]. Hex-V-I is a novel metabolite recently described by our group during a study on low dose rate effect on biofluid signatures [27]. Subsequent studies show that Hex-V-I increases the predictive performance of metabolite panels for identifying irradiated individuals across several dose rates recapitulating complex exposures encountered in realistic nuclear emergency events [60, 61]. N1-acetylspermidine is indicative of polyamine metabolism and may also be affected by age in addition to the microbiome [62]. We found that microbiome depletion increased basal levels of Hex-V-I and reduced the sensitivity and specificity for this marker to identify individuals with an 8 Gy exposure vs. the 3 Gy and sham cohorts combined. Also, as the utility of N1-acetylspermidine has not been demonstrated in nonhuman primate models, this highlights our continued efforts to integrate multiple biomarkers across time points into a single biodosimetry assay.

In the polar metabolite fraction in serum, citric acid showed the highest fold change with a dose effect observed for both Abx-con and Abx mice. While citric acid is commonly identified in urine, it has not been reported in radiation studies on serum but the trends are identical to isocitrate levels in rat serum at 1 d post-irradiation [63]. Proline betaine levels fell after an 8 Gy exposure in both Abx and Abx-con mice (but only in the urine of Abx mice), however this metabolite has been documented in the bone marrow, lung, and serum of mice post-

irradiation [64] and urine [65] of mice simply from a cage effect and due to its likely dietary source from rodent chow it was not used for further modelling or ROC curve analysis. The inclusion of carnitine into the ROC curve analysis decreased confidence intervals and increased AUROC values for all curves. Of the lipid fraction, we found several LysoPCs that decreased for both Abx-con and Abx mice, primarily after an 8 Gy exposure. LysoPCs are thought to be generally pro-inflammatory, will activate various cytokines, and will increase in an environment high in reactive species, thus would increase post-irradiation. However, several studies now show decreased serum LysoPC levels after IR exposures, especially higher doses (~8 Gy in mice). In murine models, decreased serum levels of both LysoPC (14:0) and (22:5) were incorporated into ROC curves and excellent sensitivity and specificity in wild-type mice and a p38αß$^{Y323F}$ stain used to mimic an attenuated inflammatory response, and LysoPCs were lower in rat (18:3), (20:3), (20:5) [66] (20:2), (20:3) [67] and mouse (20:3) [68] plasma post-irradiation. The level of saturation and its attached fatty acid likely play a major role in the overall function of these metabolites and their biphasic responses post-irradiation can complicate interpretation [69], however, the fact that multiple reports continue to independently identify a defined number out of the possible number of LysoPCs that can exist is intriguing.

There were of course limitations to our study. Ideally, a germ-free mouse model would be available, however, the cost of these models can be prohibitively expensive. Not all microbes will be eliminated using antibiotic treatments and there can be sex-effects as observed in the current study. Future studies may use a microbiome profiling approach to determine which microorganisms are present after this particular antibiotic regimen and how this would be different between males and females, as has been observed in a Sprague Dawley rat model [45]. Also, the use of a single metabolomics platform (here LC/MS) will not capture the whole metabolome. Using LC/MS platforms does give a more complete "snapshot" of the metabolome compared to GC/MS or NMR platforms, but some compounds may be missed. Another limitation that is beyond the scope of the current study is the incorporation of other biomarkers, such as transcripts or proteins, that may contribute to a more accurate dose reconstruction. As these studies are continually being incorporated and refined, the purpose of this study was to determine if host microbiome metabolites may be a confounder in metabolomics-based dose reconstruction.

## Conclusion

In the event of a radiation emergency from intentional or accidental exposures, biodosimetry assays should ideally provide accurate dose reconstruction irrespective of the variable demographics and genetic makeup of the individuals where it needs to be deployed. As the host microbiome is intrinsically intertwined with the resultant metabolic signature each individual will produce, here we investigated how chemically 'knocking out' the host microbiome would affect dose reconstruction using small molecule profiling of easily accessible biofluids. This approach is also relevant given that antibiotics will likely be supplied as a first defense countermeasure in a nuclear emergency. Pre-irradiation, microbiome depletion was reflected in urine by perturbation to amino acid and pyrimidine metabolism, propionic acid metabolism, hormone synthesis, and ammonia recycling, while the effects in serum were primarily observed in lipid metabolism. Post-irradiation, we found attenuated responses in mice provided with broad-spectrum antibiotics using multivariate techniques, however, several radiation markers contained in our in-house assay still gave excellent to good sensitivity and specificity for dose reconstruction. We have seen similar results in previous studies examining radiation effects in gene knockout mice that represent extreme cases of genetic variation and across varying dose

rates recapitulating realistic IR exposures. Collectively, these works provide encouraging results for the continued advancement for metabolomics based biodosimetry and in particular, their inclusion with ongoing investigations into protein and transcript radiation biomarkers.

## Supporting information

**S1 Table. Welch's t test P-value for urine metabolites that were significantly perturbed due to depletion of host microbiome.**
(DOCX)

**S2 Table. Brown-Forsythe P-values for urinary metabolites that were significantly perturbed due to ionizing radiation exposure and significant days identified by Dunnett's multiple comparisons.**
(DOCX)

**S3 Table. Mixed effects model output including the regression parameters, standard errors, p-values, and diagnostics on regression residuals.**
(DOCX)

**S4 Table. P-values for serum metabolites that were significantly perturbed due to ionizing radiation exposure.**
(DOCX)

**S1 Fig. Study design.** Urine and feces were collected from mice before they were given broad spectrum antibiotics for 8 days prior to exposure to ionizing radiation. Mice were exposed to either 0, 3, or 8 Gy ionizing radiation and urine was collected at days 0, 1, and 3 days and feces were collected at day 0 and 3. At the end of the experiment serum and tissues were collected post-mortem.
(TIF)

**S2 Fig. Base peak chromatograms.** Base peak chromatograms in ESI+ (top) and ESI- (bottom) modes of the pooled urine control sample overlaid with the NIST Standard Reference Material 3667 (creatinine in frozen human urine).
(TIF)

**S3 Fig. Base peak chromatograms.** Base peak chromatograms in ESI+ (top) and ESI- (bottom) modes of the pooled serum control sample overlaid with the NIST Standard Reference Material 1950 (metabolites in frozen human plasma).
(TIF)

**S4 Fig. DNA extraction, real-time qPCR, and urine metabolomics.** A) Heatmap produced of the top 250 urinary metabolites from male mice used for fecal DNA extraction in Table C. The heatmap shows similar profiles for mice after supplying broad-spectrum antibiotics in the drinking water in contrast to the DNA levels observed. B) Similarly, a PCA plot shows distinct differences in the urine metabolome between male mice before antibiotics vs. post-antibiotic administration. C) Table of the copy # of HV3-16S ($x10^4$) / 1 mg fecal mass for male and female mice before antibiotic administration, after antibiotic administration but before irradiation, and after antibiotic administration and irradiation. Bold text indicates statistical significance from a one-way ANOVA $P < 0.05$.
(TIF)

**S5 Fig. Contribution of the host microbiota on the mouse serum metabolome.** After removal of the host microflora, 10.2% of serum spectral features (primarily corresponding to lipids) showed significantly different levels as shown in above heatmap and volcano plot

generated from the positive mode ions in MetaboAnalyst.
(TIF)

**S6 Fig. Enrichment analysis of mouse urine post-irradiation.** The significant metabolites generated from the volcano plot of the positive mode ions was used for enrichment analysis using the small molecule pathway database (SMPD). Mice with depleted host microflora had an absence of perturbation in pathways involved in some amino acid pathways (tryptophan, glycine, serine, methionine, aspartate, arginine, β-alanine, proline), pyrimidine metabolism, propionic acid metabolism, hormone synthesis, and ammonia recycling. Perturbation to the carnitine synthesis, TCA cycle, and mitochondrial beta-oxidation were conserved between treatments.
(TIF)

**S7 Fig. Validated urinary metabolites after IR exposure.** Metabolites with statistically significant changes from radiation exposure but were not used for constructing ROC curves in the current study.
(TIF)

**S8 Fig. Validated urinary metabolites after IR exposure.** Tandem MS tables and concentrations for $m/z$ 307.2025_5.29 and 347.1227_0.27 and concentrations post-irradiation.
(TIF)

## Acknowledgments

We would like to thank the editor (Tommaso Lomonaco, Ph.D.) and anonymous reviewers for their help to improve the manuscript. We also thank the Blankenship-Paris laboratory for providing plasmids containing the HV3-16S insert (David Goulding, Dr. Blankenship-Paris laboratory, National Institute of Environmental Health Sciences). The authors would like to thank the Lombardi Comprehensive Cancer Metabolomics Shared Resource (MSR) and Center for Metabolomics Studies (CMS) for data acquisition and support.

## Author Contributions

**Conceptualization:** Evan L. Pannkuk, Heng-Hong Li.

**Data curation:** Evan L. Pannkuk, Anika Kot, Lorreta Yun-Tien Lin, Heng-Hong Li.

**Formal analysis:** Evan L. Pannkuk, Igor Shuryak, Anika Kot, Lorreta Yun-Tien Lin.

**Funding acquisition:** Evan L. Pannkuk, Albert J. Fornace, Jr.

**Investigation:** Evan L. Pannkuk, Igor Shuryak.

**Methodology:** Evan L. Pannkuk, Igor Shuryak, Anika Kot, Lorreta Yun-Tien Lin, Heng-Hong Li.

**Project administration:** Evan L. Pannkuk, Albert J. Fornace, Jr.

**Resources:** Albert J. Fornace, Jr.

**Supervision:** Evan L. Pannkuk, Heng-Hong Li, Albert J. Fornace, Jr.

**Validation:** Evan L. Pannkuk.

**Visualization:** Evan L. Pannkuk, Albert J. Fornace, Jr.

**Writing – original draft:** Evan L. Pannkuk, Igor Shuryak, Anika Kot, Lorreta Yun-Tien Lin, Heng-Hong Li, Albert J. Fornace, Jr.

**Writing – review & editing:** Evan L. Pannkuk, Igor Shuryak, Anika Kot, Lorreta Yun-Tien Lin, Heng-Hong Li, Albert J. Fornace, Jr.

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
