## [Decision Letter · Decision Letter 0]

28 Nov 2023

PONE-D-23-19621Effects of Microbiome Depletion on Radiation Biodosimetry MetabolomicsPLOS ONE

Dear Dr. Evan Pannkuk,

Thank you for submitting your manuscript to PLOS ONE. After careful consideration, we feel that it has merit but does not fully meet PLOS ONE’s publication criteria as it currently stands. Therefore, we invite you to submit a revised version of the manuscript that addresses the points raised during the review process.

We look forward to receiving your revised manuscript.

Kind regards,

Tommaso Lomonaco, Ph.D

Academic Editor

PLOS ONE

Journal Requirements:

"This work was funded by a pilot grant (P.I. ELP) from the Opportunity Funds Management Core of the Centers for Medical Countermeasures against Radiation, the National Institutes of Health (National Institute of Allergy and Infectious Diseases) grant U19-AI067773 (P.I. David J. Brenner).  The authors acknowledge the Lombardi Comprehensive Cancer Metabolomics Shared Resource (MSR), which is in part supported by Award Number P30CA051008 (P.I. Louis Weiner) from the National Cancer Institute. The authors would like to thank the Lombardi Comprehensive Cancer Metabolomics Shared Resource (MSR) for data acquisition.  The content is solely the responsibility of the authors and does not necessarily represent the official views of the National Cancer Institute or the National Institutes of Health."

6. Thank you for stating the following in your Competing Interests section:  

"No"

7. We note that you have stated that you will provide repository information for your data at acceptance. Should your manuscript be accepted for publication, we will hold it until you provide the relevant accession numbers or DOIs necessary to access your data. If you wish to make changes to your Data Availability statement, please describe these changes in your cover letter and we will update your Data Availability statement to reflect the information you provide.

8. Please include your full ethics statement in the ‘Methods’ section of your manuscript file. In your statement, please include the full name of the IRB or ethics committee who approved or waived your study, as well as whether or not you obtained informed written or verbal consent. If consent was waived for your study, please include this information in your statement as well. 

9. We note that Figure 1 in your submission contain copyrighted image. All PLOS content is published under the Creative Commons Attribution License (CC BY 4.0), which means that the manuscript, images, and Supporting Information files will be freely available online, and any third party is permitted to access, download, copy, distribute, and use these materials in any way, even commercially, with proper attribution. For more information, see our copyright guidelines: http://journals.plos.org/plosone/s/licenses-and-copyright.

Additional Editor Comments:

Dear Authors, the paper requires major revisions before to be further processed by PloSone.

Best regards,

Tommaso Lomonaco

Reviewers' comments:

Reviewer's Responses to Questions

**Comments to the Author**

1. Is the manuscript technically sound, and do the data support the conclusions?

Reviewer #1: Yes

Reviewer #2: Yes

2. Has the statistical analysis been performed appropriately and rigorously? 

Reviewer #1: Yes

Reviewer #2: No

3. Have the authors made all data underlying the findings in their manuscript fully available?

Reviewer #1: Yes

Reviewer #2: Yes

4. Is the manuscript presented in an intelligible fashion and written in standard English?

Reviewer #1: Yes

Reviewer #2: Yes

5. Review Comments to the Author

Reviewer #1: 1. In this manuscript, the authors reported their investigations on the effects of microbiome depletion on metabolomics biodosimetry. This research involves a lot of work and has novel methods. It is of great value for in dose reconstruction and novel biodosimetry assay. However, some revisions were required before publication.

2. Page 12, metabolites with FDR corrected P value of <0.10 in urine or P value <0.20 in serum were selected. Except FDR, VIP is widely accepted and expected as a standard in metabolomics analysis, why VIP was not chosen for differentially metabolites definition? And what is the reason that using different P value in urine and serum?

3. Page 15, the results of sex specific differences in microbiome depletion efficiency should be noteworthy and further discussed.

4. Fig 4. The ROC curves of metabolite panel (Six or five biomarkers) were used to classify the cluster between irradiation and control. Some biomarkers, such as citric acid and N1-acetylspermidine, have lager error bar in several groups, especially in controls. The information on individual differences of the involved metabolite should be considered and discussed.

5. Fig 5. and Fig 7. There were not enough evidences that using the ROC panel to evaluate the real accuracy rate for other blind samples, although the AUROC > 0.90, that should be noteworthy and discussed.

Reviewer #2: The authors present a well-written manuscript on the effects of microbiome depletion with regards to radiation biodosimetry metabolomics. The data are interesting but as a non-expert in the field I found it challenging to identify the research question being asked. There were a number of other areas where the manuscript could be improved to aid clarity. I also have some questions about the statistical approach and whether it is appropriate to the question being asked.

RESEARCH QUESTION

As a non-expert. this was not clear to me. I believe that it may be "Is depletion of the microbiome a confounder for radiation biodosimetry by metabolomics". If this is the case, I suggest a more positive title, rather than the current descriptive one (it is usually not a good idea to have a title that says that there is an effect, e.g. from microbiome depletion, without letting the reader know what those effects are, especially when the article concludes that there is a panel resistant to this confounder).

FIGURES

The figures are generally poorly put together and would benefit from substantial improvement.

For example, Figures 1D and 1C are too small and the labelling of the points is illegible. I cannot be sure, but it looks as if the m/z values and retention times are presented - are there no metabolite names? Whilst I appreciate that it is hard to match font sizes in multi-part figures exactly, the PCA and volcano plot have completely different axis labels, and the former are so small as to be unreadable. It does not help the reader.

As another example, Figure 2, why drop the 2A and 2B convention from the rest f the paper? It is better to be consistent. Also, the Venn diagram in Figure 2(Left, Bottom) is poor, the overlap of 241 is BY FAR the largest part and is shown as the smallest part of the Venn diagram (and also doesn't follow the colour scheme of Figure 2(Left, Top)).

Another example, Figure 3, the volcano plots are once again so small as to be pointless - what can the reader deduce from the illustrations in an efficient manner? I couldn't gain any additional information. Also, multi-part figures should each have their own lettering, it is not best practice to just say Figure 3A and include three separate charts in Figure 3A.

It is very helpful to the reader of an article, especially a non-expert, to have clearly labelled and captioned Figures

Please also be consistent, don't use Figure 3, and then Fig 6.

INTRODUCTION AND STATISTICAL APPROACH

As a non-expert I found the paper challenging to read, but I believe that the topic deals with determining radiation exposure by means of biodosimetry assays. I found that the Introduction was poor in explaining the background to this. Metabolomics is often not the first-line solution to problems, as it involves additional instrumentation. I did not understand why metabolomics was selected in preference to other tools, and would like the Introduction to better explain what UNMET NEED in the field of biodosimetry is being met here. I would have appreciated more discussion of other methods and why they are inadequate to the task, in order to understand the importance of THIS research. For example:

New Approaches for Quantitative Reconstruction of Radiation Dose in Human Blood Cells

DOI: 10.1038/s41598-019-54967-5

Biomarkers to Predict Lethal Radiation Injury to the Rat Lung

doi: 10.3390/ijms24065627

These may not be the correct articles to cite, but the Introduction must be improved to demonstrate the context and why the research is useful, there is only brief mention of previous research

Also, AUROC and PCA are not that helpful in regression problems, these are typically more useful in classification problems. If the goal is to develop a biodosimetry assay, then the appropriate statistical reporting should include assessment of regression accuracy on the sham and Abx mice, and demonstrate e.g. the RMSE, goodness of fit of the model, examine linearity. If these things cannot be done, this is a severe limitation of the study, and must be reflected in the text, because at the moment I was not at all clear that you have demonstrated that the models you have built can be used to accurately measure the dose received by the mice, and what the errors are (including error bars on the estimates, of course).

Overall, the Introduction and Discussion both need to take account of previous work, the advantages of the proposed method, and a proper statistical demonstration of the accuracy of the proposed method and reconstructing the dose with a proper assessment of the goodness of fit. Alternatively, it would be necessary to explain why PCA is preferred to other methods of assessing dose reconstruction.

Figure 6A, what is the statistical basis for the ellipses added, why are they not added to Figure 6B? Why are the ellipses in 2 dimensions on a 3D chart? Why does the green ellipse not include the outlier, whereas the red ellipse does include the outlier? If ellipses are to be added, this is best done by Hotellings or similar.

MINOR COMMENTS

Abstract Line 8 Minor comment, as a non-expert I would find "Here, we 'knocked out' the microbiome of ... mice using antibiotics (Abx mice)' more helpful.

Introduction Line 7, why have you added an acronym for IND, when IND is never again referred to in the text?

Line 15, remove the word 'incorporation' or state what these assays are being incorporated to

There is no Limitations paragraph in Discussion - this omission should be addressed.

6. PLOS authors have the option to publish the peer review history of their article (what does this mean?). If published, this will include your full peer review and any attached files.

Reviewer #1: No

Reviewer #2: No

---

## [Decision Letter · Decision Letter 1]

6 Mar 2024

Host Microbiome Depletion Attenuates Biofluid Metabolite Responses Following Radiation Exposure

PONE-D-23-19621R1

Dear Dr. Evan Pannkuk,

We’re pleased to inform you that your manuscript has been judged scientifically suitable for publication and will be formally accepted for publication once it meets all outstanding technical requirements.

Kind regards,

Tommaso Lomonaco, Ph.D

Academic Editor

PLOS ONE

Reviewers' comments:

Reviewer's Responses to Questions

**Comments to the Author**

1. If the authors have adequately addressed your comments raised in a previous round of review and you feel that this manuscript is now acceptable for publication, you may indicate that here to bypass the “Comments to the Author” section, enter your conflict of interest statement in the “Confidential to Editor” section, and submit your "Accept" recommendation.

Reviewer #2: All comments have been addressed

2. Is the manuscript technically sound, and do the data support the conclusions?

Reviewer #2: Yes

3. Has the statistical analysis been performed appropriately and rigorously? 

Reviewer #2: Yes

4. Have the authors made all data underlying the findings in their manuscript fully available?

Reviewer #2: Yes

5. Is the manuscript presented in an intelligible fashion and written in standard English?

Reviewer #2: Yes

6. Review Comments to the Author

Reviewer #2: (No Response)

7. PLOS authors have the option to publish the peer review history of their article (what does this mean?). If published, this will include your full peer review and any attached files.

Reviewer #2: No

---

## [Editor Report · Acceptance letter]

4 Apr 2024

PONE-D-23-19621R1 

PLOS ONE

Dear Dr. Pannkuk, 

I'm pleased to inform you that your manuscript has been deemed suitable for publication in PLOS ONE. Congratulations! Your manuscript is now being handed over to our production team.

Kind regards, 

on behalf of

Dr. Tommaso Lomonaco 

Academic Editor

PLOS ONE